# Synonym Expansion for Large Shopping Taxonomies

**Adrian Boteanu**                                          BOTEANUA@AMAZON.COM
**Adam Kiezun**                                             AKKIEZUN@AMAZON.COM
**Shay Artzi**                                              ARTZI@AMAZON.COM
*Amazon.com, Inc*
*Boston, MA*

## Abstract

We present an approach for expanding taxonomies with synonyms, or aliases. We target large shopping taxonomies, with thousands of nodes. A comprehensive set of entity aliases is an important component of identifying entities in unstructured text such as product reviews or search queries. Our method consists of two stages: we generate synonym candidates from WordNet and shopping search queries, then use a binary classifier to filter candidates. We process taxonomies with thousands of synonyms in order to generate over 90,000 synonyms. We show that using the taxonomy to derive contextual features improves classification performance over using features from the target node alone.We show that our approach has potential for transfer learning between different taxonomy domains, which reduces the need to collect training data for new taxonomies.

## 1. Introduction

Semantic Networks (SN) represent entities, relationships between entities, and their properties. Semantic Networks may represent a broad variety of information, from named entities, such as persons or places, to abstract concepts. The term "knowledge graph" is also used to describe this form of structured data. One of the properties commonly encoded in a SN are the primary name and aliases of an entity in multiple languages. For example, Wikidata[1] entity Q2 has multilingual names, such as Earth or Blue Planet (in English), or Tierra (in Spanish). Semantic networks may include sub-structures based on a subset of the relations defined, for example, taxonomies which define type-subtype relations; for example, ConceptNet includes the WordNet taxonomy [Speer et al., 2017] as a subset of its nodes and relations.

Synonyms, or aliases, are equivalent names for entities in a SN. For example, "washing machine" and "washer" can refer to the same concept of an appliance type. Synonyms enable improved performance in a variety of SN applications. For entity extraction from text [Cohen and Hersh, 2005, Agrawal et al., 2008], wikification [Huang et al., 2014, Cai et al., 2013], or natural language instruction grounding [Howard et al., 2014], a broader set of synonyms improves recall. In applications which use SN to generate prompts for users, such as conversational agents [Frommert et al., 2018, Yan et al., 2016] or generating explanations of the system's state in natural language [Boteanu and Chernova, 2015], a richer set of synonyms results in more varied utterances.

In this paper, we focus on the problem of expanding taxonomies with synonyms for applications in which entities are complex concepts arranged into taxonomies designed to

---

1. https://www.wikidata.org/wiki/Q2

facilitate browsing the product catalog on amazon.com. The ontologies contain product type taxonomies, which are the focus for this work, in addition to other information such as attributes for refining products in search results. In addition to distinct product types, the taxonomies contain nodes which are complex concepts, for example combinations of types and attributes, or groupings of multiple types. For example, the node "Gloves & Protective Gear" groups together gloves and other gear; the node "Automatic Irrigation Equipment" describes irrigation equipment that has automation features.

The primary application of the synonyms generated using our method is to identify direct references to the taxonomy nodes in text such as search queries. Having a broader set of synonyms for taxonomy nodes enables a broader query coverage for experiences that are specific to products in the taxonomy, for example, showing the best selling products under a given category. It is thus important to the users' experience that node synonyms are as accurate as possible, within the broader context of the taxonomy. For example, given the node "household bathroom surface cleaners" we output synonyms such as "family bathroom surface cleaner" and "house bath surface cleansing." Our method is robust to errors of word sense compatibility, for example we reject "mack game restrainer" as a synonym for "mac game controllers," or "store circuit board" is a rejected candidate for "memory cards."

The taxonomies are authored by experts familiar with the respective shopping domains to facilitate navigation and browsing (Section 4.1). They contain over 4,300 nodes and have depths of over 30 nodes; in addition to taxonomical relationships, they represent type properties, possible values, node equivalence, and other information. In this paper, we identify each taxonomy by its root node name. For the example shown in Figure 1, the taxonomy "Baby Products" includes, among 15 other nodes, a category node named "Car Seats and Accessories." This has the children "Car Seats," "Car Seat Bases," "Car Beds," and "Accessories." The "Accesories" node has 17 children (e.g. "Cup Holders" and "Seat Liners"), while the "Car Seats" node has five children grouped by age group and chair type. We note the fine granularity of nodes, which includes distinctions based on product types, features, indented use, and other criteria dependent on the domain; concepts range from general to specific in fine increments, with children refining and specifying the parent node. The taxonomy nodes we target have complex names, for example "Convertible Child Safety Car Seats" and are thus unlikely to be frequently found in large natural language text corpora with sufficient frequency in order to extract synonyms from unstructured text.

We present a method that leverages similarity within the taxonomy to evaluate synonym candidates obtained using low-precision, high-recall methods. Our goal is to enable collecting possible synonyms from a broad range of sources, and output a final set of synonyms consistent to a single standard. This method enables expansion with synonyms for complex SN that are not common in typical text corpora, such as shopping taxonomies for browsing. The main advantages of our approach are that: 1) it does not depend on frequent mentions in corpora of entities in the taxonomy; 2) it identifies synonyms that fit within the broader structure of a taxonomy contained within the graph, and outputs synonyms of similar specificity to the original name; 3) the classifier uses domain-independent features, enabling cross-domain predictions.

Our method consists of the following stages (Figure 2):

1. **Generate synonym candidates** for each node of the taxonomy. We experimented with two methods of candidate generation. First, we primarily used a method based on

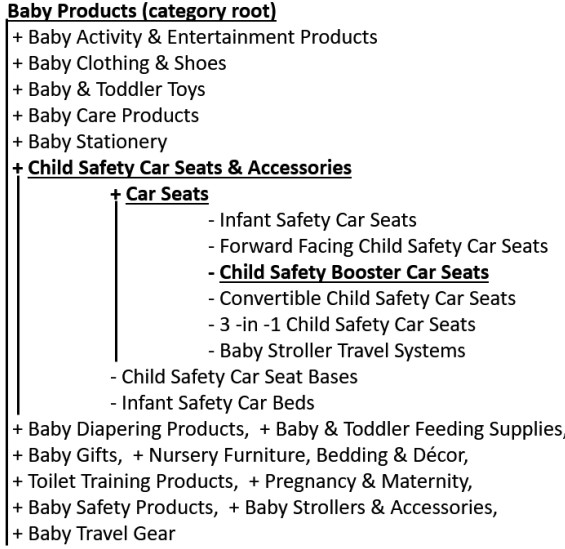

**Baby Products (category root)**
+ Baby Activity & Entertainment Products
+ Baby Clothing & Shoes
+ Baby & Toddler Toys
+ Baby Care Products
+ Baby Stationery
+ **Child Safety Car Seats & Accessories**
      + **Car Seats**
            - Infant Safety Car Seats
            - Forward Facing Child Safety Car Seats
            - **Child Safety Booster Car Seats**
            - Convertible Child Safety Car Seats
            - 3 -in -1 Child Safety Car Seats
            - Baby Stroller Travel Systems
    - Child Safety Car Seat Bases
    - Infant Safety Car Beds
+ Baby Diapering Products,  + Baby & Toddler Feeding Supplies,
+ Baby Gifts,  + Nursery Furniture, Bedding & Décor,
+ Toilet Training Products,  + Pregnancy & Maternity,
+ Baby Safety Products,  + Baby Strollers & Accessories,
+ Baby Travel Gear

Figure 1: Sample section of a taxonomy used in this work, which is designed for exploring and filtering online shopping catalogs. We highlight the path from the root node, "Baby Products," to a leaf node, "Child Safety Booster Car Seats." Each node prefixed by a + sign indicates the node has children; leaf nodes are marked by a -. For compactness, we enumerate instead of indenting some of the 15 children of the root node.

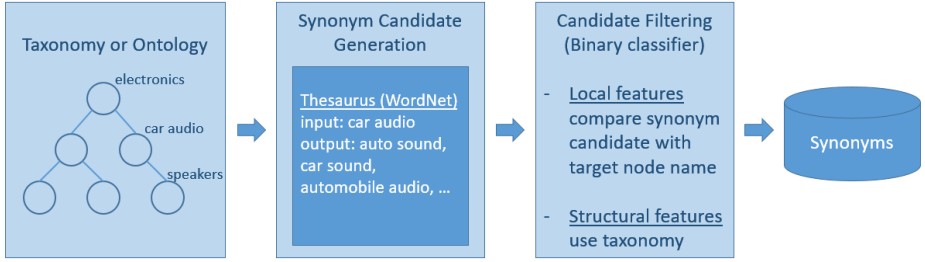

Figure 2: Overview of our method. We start with product taxonomies designed for browsing a large online shopping catalog, described in Section 4.1, and generate synonym candidates for each node using a thesaurus such as WordNet (Section 3.1). We then classify the set of candidates using a binary classifier (Section 3.2) to output the final set of synonyms.

WordNet [Miller, 1995], to generate the cartesian product of concept-level synonyms that are present in the node's name (Section 3.1). Secondly, we show additional results on classifying shopping search queries (Section 4.4).

2. **Filter synonym candidates** using a binary classifier (Section 3.2). The classifier uses features derived from a) similarity between the candidate the target node, and b) similarity features between the candidate and other nodes in the taxonomy. Our goal is to avoid producing synonyms more general or more specific than the original node name, such that the synonyms are consistent with the taxonomy as a whole. The classifier uses features independent of the taxonomy vocabulary, making our method suitable for transfer learning by predicting on new taxonomies that do not have training data available. Transfer learning is one method of interest to reduce the need to collect training labels for new taxonomies.

The rest of the paper is structured as follows. We first review relevant literature. We then describe the taxonomies we use in this work (Section 4.1), and the methods of obtaining synonym candidates and classifying them. We then evaluate the binary synonym classifier using a corpus of annotations collected using crowdsourcing for synonyms generated using the thesaurus. We also include cross-domain learning experiments to evaluate the potential for training the classifier on one taxonomy and predicting on synonyms for different taxonomy (Section 4.3). Furthermore, we conducted a separate evaluation using an alternative method of selecting synonym candidates, which we will briefly summarize: we associated search queries with taxonomy names using customer purchases, and used these search terms as synonym candidates (Section 4.4). We evaluate the impact of using domain-specific knowledge, specifically lists of known brand names, which may be closely associated but not synonymous with product categories, to improve synonym filtering. We conclude the paper with observations about the role of taxonomy-wide similarity in predicting synonymy and describe future directions.

## 2. Related Work

Methods of automatically buiding knowledge bases from unstructured text include identifying salient terms (keywords and keyphrases), identifying synonymy between terms, forming entities and entity hierarchies from keywords, and inferring relationships and rules between entities [Buitelaar et al., 2005]; this covers the spectrum from keyword extraction to semantic inference. Synonym extraction has been defined as one of the base steps of automated ontology and knowledge base building.

Synonym expansion has been used to enrich search keywords in order to improve recall. WordNet synonym sets have been used to expand search keywords [Voorhees, 1994]. This work used rules on the WordNet taxonomy to expand queries with keywords, increasing the number of words that are matched. Other work has used text mapping to concepts in a thesaurus for query expansion [Aronson et al., 1994]. In life sciences, domain-specific ontologies have been used in similar ways to expand search terms [Yunzhi et al., 2016].

Previous work in extracting synonyms from text relies on frequent mentions of the entities in text corpora [Cho et al., 2017, Leaman and Lu, 2016, Yates et al., 2014, Henriksson et al., 2014]. These works identify synonyms based on statistically similar contexts or

phrases in which they are used. The taxonomies we use in this work are engineered to facilitate exploring a product catalog for online shopping, for example by choosing subtypes of products or filtering by feature. As such, we cannot expect that all node names and their potential synonyms will occur in text. Other work has used clustering methods that identify synonyms such as "birth date" and "date of birth" in search queries [He et al., 2016]. Topic-based methods have also been used in medical document search [Huang et al., 2017] to add aliases as encountered in text. We show an approach that uses search queries and past customer purchases to propose synonym candidates (Section 4.4); this method benefits from using domain-specific knowledge, such as product brand names.

Other work in identifying aliases has been focused on named entities, as opposed to common words. Named entity recognition and extraction (NER) are rich research areas, focusing on identifying and categorizing proper names in text [Etzioni et al., 2005, Nadeau et al., 2006, Mohit, 2014, Habibi et al., 2017]. Named entities may refer to persons, places, brands, or authors. It is a problem related to synonym detection since the same person may be referred to in different contexts using different names. The problem we address in this paper is different than existing work in NER and synonym extraction from unstructured text. Our entities cannot be considered names entities, and as mentioned above, we do not expect to find them mentioned frequently in text corpora.

Structure mapping is an established approach of comparing semantically-complex structures, with applications in analogy modeling [Gentner and Markman, 1997, Markman and Gentner, 1993, Falkenhainer et al., 1989, Forbus et al., 2017]. Structural similarity methods consider correspondence at a relational level to be indicative of higher similarity, as opposed to feature-based similarity models that compare attributes directly. For example, in a structural similarity approach, the function an entity performs is more important than individual features such a color. Previous work has used structural similarity in contexts derived from robot tasks in order to identify equivalent objects for a given task [Boteanu et al., 2015]. Other work using structural similarity for comparison identified significant discrepancies in superficially similar structures, for example to make the distinction between an arch and a bridge [McLure et al., 2015]. In designing the synonym filtering classifier, we incorporated concepts from structure mapping by adding features that compare the synonym candidate with multiple nodes in the taxonomy; we refer to these as structural similarity features. The rationale behind this decision was to calibrate the level of generality of a synonym by considering the surrounding nodes of the taxonomy.

## 3. Method

Our approach consists of two stages. First, we identify synonym candidates. We describe a method based on a thesaurus containing concepts part of a node's primary name (Section 3.1). Second, we filter synonym candidates using a classifier (Section 3.2). The rationale behind our design is that multiple methods can be used to generate candidates; we demonstrate a generally-applicable method. The filtering stage processes candidates regardless of the method used to obtain them.

### 3.1 Candidate Generation

We start from the observation that many taxonomy node names are composed of common words for which we could identify synonyms. For example, the node "Baby Clothing & Shoes" has the synonym "Baby Clothing & Footwear." One of the challenges in this approach is selecting word-level synonyms that are consistent with the sense of the original word. We identify word synonyms using WordNet [Miller, 1995]. WordNet represents concepts as *synonym sets*, or synsets, arranged in a taxonomy. Each synset consists of synonyms for a concept corresponding to a particular sense of the word. For example the word "car" may have the meanings Car.n.01: car, auto, automobile; Car.n.02: car, rail car, railroad car; Car.n.03: car, gondola.

For each node in the taxonomies:

1. We split the node's name into separate concepts found in WordNet. We combine individual consecutive words into concepts whenever possible.
2. We select all synsets for the words in the node name.
3. We perform word sense disambiguation in order to find the WordNet synset to the taxonomy. We select the WordNet synset most similar to the node and its context for each concept identified in the node name. We define context as the node name and the other node names in the taxonomy. We compute similarity by averaging cosine similarity over word embeddings between all pairwise concept pairs between the synonym set and the node context. We choose the synset that results in the highest average similarity to use for permuting word synonyms. The majority of words in our taxonomies have more than one synset in WordNet, we designed this step to prune the set of incorrect candidates.
4. We generate the cartesian product of all synset words extracted from WordNet for each concept in the node's name.

The following is an example:

1. Given a node name "washing machine parts," we identify the corresponding concepts, "washing_machine" and "parts."
2. We build a context vocabulary by sampling other node names from the Electronics taxonomy. We compare all synsets corresponding to "washing_machine" and "parts" to the context vocabulary. We select the most similar sense for each concept, "washer.n.03" and "part.n.01."
3. We extract all lemmas for each synset, i.e. "washer," "automatic_washer" and "washing_machine," and "part," "portion," "component_part," "component."
4. We generate the cartesian product of these lemmas starting from the original node name, for example "washer parts" or "automatic_washer components," for a total of 12 phrases. Each resulting list of lemmas is a synonym candidate, which is then accepted or rejected using the method described below.

### 3.2 Candidate Filtering

We select candidates that 1) have similar meaning to the original node name, and 2) have a similar level of generality with the original name node, in the context of the taxonomy.

In common speech, the product types "TV" and "LED TV" may be considered equivalent, with most shoppers referring to an LED TV as simply TV, assuming that LED is the most common display technology at a given time. However, from our taxonomy standpoint they are not synonyms: the type TV has other sub-types, such as OLED TV, Plasma TV, CRT TV, and LED TV. We hypothesized that, in order to make such distinctions when identifying synonyms, taking into account the structure of the taxonomy is an essential factor.

We use vector word representations, or word embeddings, to compute some of the classifier's features by comparing synonym candidates with various node names in the taxonomy. To compute features, we sum the corresponding vectors for each word in a taxonomy node name or the synonym candidate(using bi- and tri-grams when available); then, we compute the cosine similarity value between the resulting addition vectors. We used Numberbatch, a set of word embeddings generated the ConceptNet semantic network, because it is publicly available, it includes WordNet concepts, and has low bias [Speer et al., 2017]. We implemented a binary gradient boosting classifier using the Python scikit-learn tookit [Pedregosa et al., 2011].

Many entity names and synonym candidates consist of more than one word, and have corresponding embeddings, such as *washing_machine*. We identify multi-word concepts by searching WordNet with adjacent n-grams in decreasing order of length, in a greedy approach that extracts the longest match from the node name. Synonym candidates generated using WordNet similarly may consist of multi-word concepts as synonyms for single words. We compute the word embedding distances between node names and synonym candidates by first summing all respective vectors for the node and the synonym candidate, then computing cosine similarity between the resulting vectors.

We group features in local and structural, where local features are computed only with respect to the target node, and structural features refer to other nodes in the taxonomy.

- **Local Features:**

  - Word frequency in search queries: We compute four features, for the average and minimum frequency of words in the original node name and in the synonym string. We denote this group of features $WF$;

  - Character and word edit distance: We compute two features from edit distance in character and in words between the candidate and the original name $(edit(SC, N))$;

  - Cosine similarity: We compute one feature by summing the word embedding vectors of the synonym candidate words $(SC)$, and comparing this sum with the word embedding sum of the original node name $(d(SC, N))$.

- **Structural features:** We compute word embedding similarity between the synonym candidate and the following, and add one feature for each (examples refer to nodes shown in Figure 1):

  - The node's parent name, e.g. "Child Safety Car Seats & Accessories" for "Car Seats" $(d(SC, P))$

  - The name of the taxonomy root, "Baby Products" $(d(SC, R))$.

– The average distance to the node's direct children ($\bar{d}(SC, ci)$, where $ci \in N.children$), if the node has children, e.g. all nodes including "Infant Safety Car Seats," "Forward Facing Child Safety Car Seats," and "Baby Stroller Travel Systems."

Our hypothesis, which we test in Section 4.3 using feature ablation, is that comparing a synonym candidate with the broader structure of the taxonomy models how specific a node is with respect to others.

## 4. Experiments

We first describe the taxonomies we used in the evaluation (Section 4.1). We then describe the methodology for collecting training data for training classifiers and show results for classifying synonyms generated using WordNet, exploring feature contribution and transfer learning between taxonomies (Sections 4.2 and 4.3). We include separate results of predicting synonyms from search queries and the effect of using domain-specific features (Section 4.4).

### 4.1 Product Shopping Taxonomies

We use taxonomies created by manual ontology design and aimed to enable navigation through the amazon.com product catalog. The taxonomies enable users to refine products by category and features, for example by selecting a product type after performing a search. We used the following eight taxonomies, with total number of nodes shown in parentheses, which we selected to cover a broad range of product types:

- Clothing, Shoes & Jewelry (4370 nodes)
- Sports & Outdoors (3997 nodes)
- Home & Kitchen (2446 nodes)
- Electronics (1714 nodes)
- Patio, Lawn & Garden (919 nodes)
- Baby Products (621 nodes)
- Beauty & Personal Care (581 nodes)
- Pet Supplies (549 nodes)

The taxonomies contain type nodes on the leaves, for example "Bulb Planters," and grouping nodes at higher levels, for example "Gardening & Lawn Care" is a grouping node with children such as "Hand Tools," which in turn has children such as "Picks," "Bulb Planters," or "Manual Lawn Aerators." Figure 3 shows a histogram of the number of words in node names in our taxonomies; for example, the node name "women's contemporary clothes designer base layer sets" has seven words. The majority of nodes have three or more words in the name. The taxonomies contain 2379 distinct words.

### 4.2 Crowdsourced Label Collection

We generated candidates using the WordNet method described in Section 3.1. For the total of 15,197 nodes in the eight taxonomies, we generated 182,974 synonym candidates. We took a uniform sample, and labeled them using a crowdsourced survey[2]. We sampled uniformly

---

2. We used Amazon Mechanical Turk, www.mturk.com

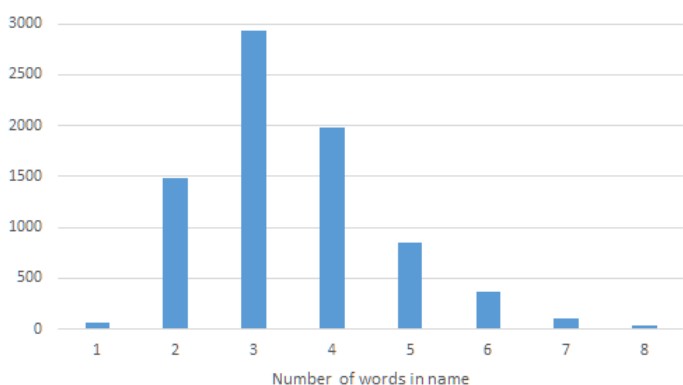

Figure 3: Histogram of words per unique node name or synonym candidate.

3331 nodes with distinct names and a total of 4488 corresponding synonym candidates for the eight taxonomies. The candidate synonym set contains 3056 distinct words (0.91 distinct words per node). We designed the survey to present the original node name, a synonym candidate, and the category name. We chose the context as the name of the root in each taxonomy tree, for example "Electronics." Participants were asked to answer a binary yes/no question, "Do these phrases mean the same in the following context?" We collected 10 answers from separate participants for each name—synonym candidate pair. We calculated the proportion of "yes" answers and used a threshold, which we refer to as the *annotation threshold*, to assign the synonym example a final label, positive or negative.

We used all eight taxonomies and collected ten answers per question, totaling 45,000 responses. Figure 4 shows the proportion of positive answers per evaluation. We observe a skew towards positive answers, which is to be expected given that the generation method incorporates word sense disambiguation to reduce the number of implausible candidates. We use this label set in Section 4.3, and explore the effect of choosing an annotation threshold for this label set.

### 4.3 Evaluation of Candidate Filtering

We trained the classifier on 90% of the labeled data and tested it on the remaining 10%. We supplemented each instance in the training set with ten negative examples by selecting names of other nodes in the taxonomy; for example, in the Electronics taxonomy, we provided examples such as "mp3 player" and "televisions" as a negative synonym example for "LED TVs." We trained and tested using this process 50 times for each condition, and report average precision and recall values.

We explored the effect of the annotator agreement threshold at which the annotation is considered positive; for this set of experiments we enabled all features of the classifier. Table 1 shows classification performance for different annotation thresholds and train/test splits in the Electronics taxonomy. For the remained of the experiments using crowdsourced labels, we select an annotation threshold of 0.6, as it achieved high precision (our primary consideration) and satisfactory recall.

We observed that using a combination of local and structural features results in the best performance (0.92 F-1 score), compared to local features or structural features alone

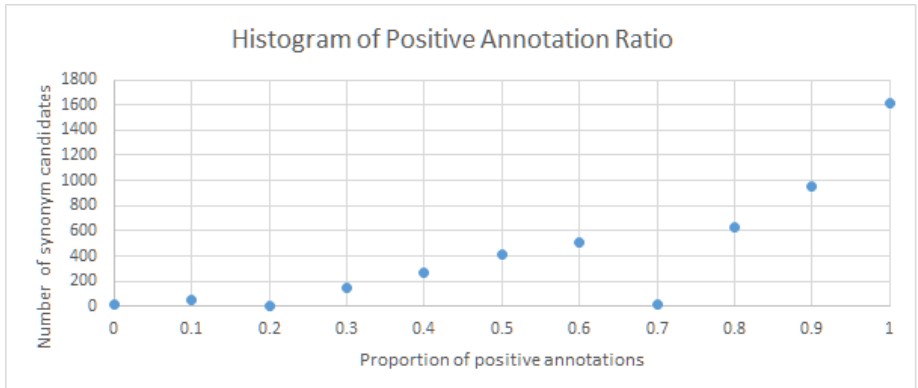

Figure 4: Distribution of positive responses from collecting synonym candidate annotations via crowdsourcing (45,000 survey responses in total, 10 per synonym candidate). There are 12 distinct candidates with 0 positive answers, and 9 distinct candidates with 7 positive annotations each (out of 10 in total).

| Annotation Threshold | Crowdsourced Positive Labels Count | Precision | Recall | F-1 |
|:---:|:---:|:---:|:---:|:---:|
| 0.4 | 431 | 0.92 | 0.75 | 0.82 |
| 0.5 | 431 | 0.92 | 0.75 | 0.82 |
| **0.6** | 398 | **0.92** | **0.79** | **0.85** |
| 0.7 | 356 | 0.88 | 0.80 | 0.84 |
| 0.8 | 299 | 0.72 | 0.64 | 0.68 |
| 0.9 | 200 | 0.67 | 0.50 | 0.57 |

Table 1: Classification performance for Electronics taxonomy for various thresholds of inter-annotator agreement. Resulting positive labels count shown out of a total of 519 synonym candidates labeled via crowdsourcing (10 annotations per candidate). Total training set included, for each node, 10 sampled negative examples for other taxonomy nodes. We selected the threshold of 0.6.

| Row | Local Features | | | Structural Features | | | Metrics | | |
|---|---|---|---|---|---|---|---|---|---|
| | $WF$ | $edit(SC, N)$ | $d(SC, N)$ | $d(SC, P)$ | $\bar{d}(SC, ci)$ | $d(SC, R)$ | P | R | F-1 |
| 1 | x | x | x | x | x | x | 0.92 | 0.79 | 0.85 |
| 2 | x | x | x | | | | 0.87 | 0.73 | 0.79 |
| 3 | | | | x | x | x | 0.90 | 0.41 | 0.56 |
| 4 | x | x | x | x | x | | 0.94 | 0.90 | 0.92 |
| 5 | x | x | x | x | | x | 0.94 | 0.90 | 0.92 |
| 6 | x | x | x | | x | x | 0.94 | 0.88 | 0.91 |
| 7 | x | x | | x | x | x | 0.95 | 0.67 | 0.79 |
| 8 | x | | x | x | x | x | 0.94 | 0.89 | 0.92 |
| 9 | | x | x | x | x | x | 0.94 | 0.89 | 0.91 |
| 10 | x | x | x | | x | | 0.95 | 0.87 | 0.91 |
| 11 | | | x | | | | 0.87 | 0.70 | 0.77 |
| 12 | x | | | | | | 1.0 | 0.07 | 0.13 |

Table 2: Classification performance (precision, P, recall, R, and F-1 score) for the Electronics taxonomy for classifiers using subsets of the features described in Section 3.2. We observe that including some structural features, by comparing synonym candidates with the root, parent or children of the target node, improves classification accuracy.

(0.56 and 0.79 respective F-1 scores). We conducted an ablation study to investigate the contribution of each category of feature. We disabled different combinations of features and trained separate classifiers for each subset of features. For brevity, we report representative findings in the Electronics category (Table 2). We note that in isolation, either local features (row 2) or structural features (row 3) result in lower recall and precision, with combinations of the two resulting in improved performance. The cosine similarity between the candidate between the target name and the candidate is important: removing it decreases recall to 67% (row 7), and on its own this feature achieves an F-1 score of 0.77 (row 11). Including a mix of local and structural features yields in similar performance of 94% precision and a 89-90% recall (rows 4, 5, 8, 10). We observe lower F-1 scores when using all three structural features, and improvements when selecting two out the three. Frequency-based features have high precision but low recall (row 12), which is expected since, if the keywords are present in large volume in search queries, it is likely that they are meaningful phrases. The thesaurus-based method generates invalid candidates primarily because of the individual word senses do not have the same meaning when put together, which means that the combination will have lower search term frequency. However, it does not identify less frequent terms, which is where in practice synonyms are most useful for, since they allow interpreting infrequent queries.

We observed similar performance in the other taxonomies, with F-1 scores of over 0.83 (with the exception of the "Clothing, Shoes & Jewelry). We selected the feature set on row 4 in Table 2, i.e. all features except for $d(SC, R)$, and evaluated prediction performance in all other categories. Table 3 shows the best performance for each of the eight product type taxonomies. We note similar scores, with the exception of Clothing, which had a significant

| Taxonomy | Precision | Recall | F-1 |
|---|---|---|---|
| Clothing, Shoes & Jewelry | 0.97 | 0.54 | 0.70 |
| Home & Kitchen | 0.90 | 0.90 | 0.90 |
| Baby Products | 0.93 | 0.75 | 0.83 |
| Electronics | 0.94 | 0.90 | 0.92 |
| Beauty & Personal Care | 0.96 | 0.85 | 0.90 |
| Sports & Outdoors | 0.88 | 0.80 | 0.84 |
| Patio, Lawn & Garden | 0.94 | 0.90 | 0.92 |
| Pet Supplies | 0.94 | 0.89 | 0.91 |

Table 3: Classification results for each product taxonomy, using 10% of annotations for testing; showing average values for 50 train/test samples. We used the feature set shown on row 4 in Table 2.

skew towards positive annotations. Including the candidate-root similarity features resulted in lower performance, similar to our previous observation (average F-1 score of 0.78, lower than the average of 0.86 shown in Table 3).

Finally, we conducted cross-domain experiments and observed degraded but comparable classification performance to using the same taxonomy for training and testing. We trained a classifier using one taxonomy and used it to predict in another. This is possible because none of the features described in this section rely on domain-specific information such as known word-level labels. We experimented with all 56 combinations of source-target combinations. For this set of experiments, we used the full feature set (i.e. not excluding $d(SC, R)$, since it resulted in a higher average F-1 score.

In Table 4 we show results, in order of F-1 score, of the top 5 best performing and top 5 worst performing from all pairwise combinations of taxonomies; for the sake of brevity we exclude the full list. The average F-1 score over all 56 combinations was 0.766 with a standard deviation of 0.087. We observe that for all top five worst performing, the target taxonomy is Clothing, Shoes & Jewelry; this was the taxonomy with the lowest score in our in-domain experiments (Table 2). The low F-1 scores are due to recall (0.35), precision is competitive (0.94). Similarly, for the top performing target taxonomies Electronics is the most common: precision and recall are both similar but slightly lower on average (0.90 precision compared to 0.94 in-domain, and 0.84 recall compared to 0.90). The prediction from Electronics to Clothing, not shown in the table, has an F-1 score of 0.67. We attribute these changes in performance to label noise, but also to how diverse the taxonomies are.

### 4.4 Synonym Candidate Selection from Search Queries

In this section we describe additional results using a separate source for synonym candidates. We experimented with selecting synonym candidates from search queries on amazon.com. These associations are derived using product purchases that follow search queries, and using the taxonomy associations for these products [Peery and Nguyen, 2017]. We reference a related method, also applied to an online shopping domain, of associating product attribute values indexed in a shopping catalog [Wu et al., 2017]. We use user behavior such as

| Source Taxonomy (Train) | Target Taxonomy (Test) | Precision | Recall | F-1 |
|---|---|---|---|---|
| Home & Kitchen | Clothing, Shoes & Jewelry | 0.94 | 0.35 | 0.51 |
| Baby Products | Clothing, Shoes & Jewelry | 0.95 | 0.35 | 0.52 |
| Beauty & Personal Care | Clothing, Shoes & Jewelry | 0.94 | 0.39 | 0.55 |
| Sports & Outdoors | Clothing, Shoes & Jewelry | 0.95 | 0.42 | 0.58 |
| Patio, Lawn & Garden | Clothing, Shoes & Jewelry | 0.96 | 0.47 | 0.63 |
| Clothing, Shoes & Jewelry.txt | Pet Supplies | 0.88 | 0.82 | 0.85 |
| Baby Products | Electronics | 0.91 | 0.81 | 0.86 |
| Pet Supplies | Electronics | 0.90 | 0.84 | 0.87 |
| Sports Outdoors | Electronics | 0.91 | 0.85 | 0.88 |
| Clothing, Shoes & Jewelry | Electronics | 0.88 | 0.89 | 0.89 |

Table 4: Classification performance for cross-domain performance, in which we trained on one taxonomy and predicted on another. The table shows the lowest and highest five pairs of source and target taxonomy, ranked by F-1 score. We used the feature set shown on row 1 in Table 2; the full set of features resulted in the highest average F-1 performance.

product purchases after issuing queries to infer statistical associations between a query and the product; we then use the product's assignment to taxonomy nodes to infer associations between the query and the node. This model associates unique search keyword queries to taxonomy nodes, and outputs a probability distribution over the taxonomy. We selected as synonym candidates queries that occurred frequently, at least once every day for an entire month, and that had a probability of over 80% to lead to purchases from the respective taxonomy node.

The following are examples of node names in the Electronics taxonomy, followed by a sample of queries selected for synonym candidates; we selected examples that are not covered by the thesaurus method:

- Portable cell phone power banks: "battery bank"
- Cell phone cases: "iphone 4s case," "lg g4 phone case"
- Repeaters: "wifi repeater"
- Hdmi cables: "hdmi to mini displayport cable"

While this method has the advantage of accessing a broad and current vocabulary, we also observe a mix of brands, product models, and manufacturer names in those categories; for the task of generating synonyms, these examples are undesirable since they are specific to a subset of items under the taxonomy node, and not to the node globally. This method has the potential of identifying candidates that are not reformulations of the taxonomy nodes. For example, the taxonomy node "Self-Balancing Scooters" may be referred to as "hover boards" in search queries. Using information from search queries enables us to identify emerging synonyms before this information is included in a thesaurus such as WordNet.

### 4.4.1 Search Query Candidate Classification Evaluation

The queries selected using this method have high lexical variability as a result of the variety of intents of search queries. For example, queries for a specific product that is assigned to a single taxonomy node with result in a high probability of the query targeting the corresponding node, because the query is unlikely to lead to clicks or purchases other than for the target product. This makes the input vocabulary to the classifier more varied, both in unique words and in the generality of those words; some candidates contain tokens such as model numbers and brands.

Using the same methodology as described in Section 4.2, we collected annotations for a uniform sample of 2000 candidate search queries for 337 nodes in the Electronics taxonomy, also with 10 answers per candidate (20,000 answers in total). The 2000 synonym candidates contain 1307 unique words. The ratio of unique candidate words per node is 3.87, 300% higher than the ratio of unique words in original node names per node, of 1.27. The synonym candidate vocabulary is significantly more diverse than for WordNet-generated candidates: for 594 nodes use for collection in Electronics in Section 4.2, there were 1.41 distinct words per node for synonym candidates, only 20% higher than the ratio of 1.17 distinct words in original names per node. Similar to the WordNet survey, we observed a skew towards positive answers. In addition, these annotations showed more positive responses for popular brands in a given product category. We evaluate using domain-specific features, such as known brand names.

For candidates that included names of manufacturers or brands, we observed a higher proportion of positive answers for brands that are more common in their respective product category, similar to genericized brands. To control for this effect, we collected an additional set of annotations, authored by experts familiar with product search. For the experiments in this section, we used a set of unigram and bigram embedding vectors generated by applying the word2vec algorithm on the search query dataset [Mikolov et al., 2013]. We used the this set of embeddings as they represent brand names in relation to a rich vocabulary, which we evaluate towards the end of this section. Following the same methodology as in Section 4.3 we trained and tested separate classifiers using the crowdsourced or the expert annotations, using all available features.

We included a feature in the model, $\beta$, that activates when a brand, product line, or manufacturer, is present in the synonym candidate. The list of brands is domain-specific, for example "apple" would be a brand name in Electronics but not in Grocery. We computed $\beta$ using a subset of 120 brands and manufacturer names extracted from the product catalog.

Table 5 shows the effect of the annotation consensus threshold and of using the feature $\beta$ using the crowdsourced annotations and when training on the expert-annotated candidates. In all these experiments, we used the full feature set described in Section 3.2 and the same setup as in the previous section. Overall, we observe a decline in recall compared to classifying performance on generated synonyms, which we attribute to the greater lexical variety in the valid candidates set; precision is maintained or improved upon compared to candidates generated with WordNet.

Classification performance is lower for the expert annotations than for the crowdsourced annotations. Furthermore, using $\beta$ is beneficial only for the expert annotation set. We attribute these observations to the effect of genericized brands. Annotations diverge between

| Annotation threshold | Using $\beta$ | Precision | Recall | F-1 |
|:---:|:---:|:---:|:---:|:---:|
| 0.6 | no | 0.96 | 0.35 | 0.52 |
| 0.6 | yes | 0.97 | 0.36 | 0.52 |
| 0.7 | no | 0.98 | 0.37 | 0.53 |
| 0.7 | yes | 0.99 | 0.36 | 0.53 |
| 0.8 | no | 0.86 | 0.34 | 0.49 |
| 0.8 | yes | 0.87 | 0.34 | 0.49 |
| expert annotation | no | 0.94 | 0.26 | 0.41 |
| expert annotation | yes | 0.97 | 0.29 | 0.45 |

Table 5: Classification performance when training with crowdsourced labels, for the Electronics taxonomy for candidates selected from search query keywords. We evaluated setting an annotation consensus threshold and a domain-specific feature, $\beta$, which identifies the presence of brand names in the candidate. Last two rows show classification performance when using expert annotations.

crowdsourced and expert annotations for categories in which popular brands are closely identified with the product category.Word embedding vectors trained without domain supervision place those words close to common words denoting the type, since they occur in the same context, making the problem inseparable for the classifier. Introducing domain knowledge, in the form of the known brand feature $\beta$, is useful only if the annotation set is free of this conflation between brand and type.

## 5. Conclusion

Entity aliases are an important component of ontology construction, enabling entity recognition in text and generating natural language references to entities. We demonstrate a method for identifying synonyms for large taxonomies used for online shopping. Our method consists of two complementary approaches of selecting synonym candidates, and a candidate filtering stage which uses a classifier that includes structural similarity features. We show that using structural similarity features, such as comparing synonym candidates with the parent, children, or root nodes in the taxonomy, improves classification accuracy, and that the method is applicable to transfer learning between taxonomies. We include an additional evaluation on using search queries associated statistically with the taxonomy nodes via user behavior. This method extracts a broader vocabulary for the candidates, including tokens that are not common words, such as proper names, model numbers, or years. We show that using domain knowledge such as brand name definitions improves classification performance for candidates extracted from search queries, which conflate in the same context types, brands and other terms.

In future work we will experiment with taxonomies in languages other than English. We will explore the potential for predicting synonyms in other languages than the training language, similar to the experiments we showed for cross-domain prediction.

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
