# OpenReview forum: "Synonym Expansion for Large Shopping Taxonomies"
_AKBC.ws/2019/Conference — AKBC 2019_

### Official Review · AnonReviewer2 · 2019-01-05
**Simple approach for a practical use case**

**Rating:** 6
**Confidence:** 4

**Review:**

The authors describe and evaluate two approaches to collecting alternative aliases (synonyms) for entities in a taxonomy: expansion from WordNet synsets and from search queries followed by a binary classification to refine the generated candidate sets. Mitigating vocabulary mismatch in search applications provides a good motivating use case for ontology/taxonomy construction and is an important research direction.

Questions:
* How were the negative samples for training the classifier selected in 4.3?
* What is the overlap between the synonym sets generated using WordNet and the search queries?
* Can the WordNet-generated candidates improve performance for aligning synonyms collected from search queries? i.e. output of the first method as input to the second synonym selection method.
* Are there other evaluation results that can show improvement from implementing the proposed approaches on the target tasks, e.g. search or information extraction?

Remarks:
* Semantic Network seems to be a synonym for a Knowledge Graph, which is a more frequently used term. The relation has to be made explicit.
* The structure of the paper is confusing: only one of the candidate selection methods is described in the Section 3 but experimental results for two approaches are reported in Section 4.

---

### Official Review · AnonReviewer1 · 2019-01-09
**Interesting approach to taxonomy extension**

**Rating:** 7
**Confidence:** 3

**Review:**

The paper presents an interesting approach to taxonomy extension that is based on identifying synonyms for component words of multi-word terms in the taxonomy. The approach seems to rely very much on WordNet, which may be a weakness. Coverage of WordNet is rather limited and the approach may therefore be limited in application. Also, word sense disambiguation (selecting the appropriate sense for a component word) is a challenge that has not been addressed in full detail, although this will be dealt with by the classification step in filtering, if I understand correctly. Overall, the paper is well-written and clear in ambitions and achieved results. The experiments use an extensive crowdsourced gold standard, which is a valuable research outcome on its own if it will be released publicly.

---

### Official Review · AnonReviewer3 · 2019-01-10
**Authors propose a two stage synonym expansion framework for large shopping taxonomies. Paper is well written and empirical study is well conducted.**

**Rating:** 7
**Confidence:** 5

**Review:**

Authors present a method for expanding taxonomies with synonyms or aliases. The proposed method has two stages, 1) generate synonym candidates from WordNet and then 2) use a binary classifier to filter the candidates. The method is simple and effective. Paper is well written with ample empirical study and analysis. Couple of minor comments:
1) Rather than using WordNet, for step 1, is it possible to use a similarity based clustering method to mine candidates for each concept from a corpus?
2) For the word embeddings used in step 2, did the authors use off-the-shelf pre-compuated embeddings or compute the embeddings from a domain specific (in this case shopping) corpus? Will the performance improve if a domain specific embedding is applied?

---

### Meta-Review · Area_Chair1 · 2019-02-11
**Interesting approach to taxonomy expansion**

**Recommendation:** Accept (Poster)
**Confidence:** 3

**Metareview:**

The work provides an interesting and yet rather straightforward approach to synonym expansion that relies on a combination of user queries and existing background knowledge in terms of WordNet. The evaluation shows good results for an interesting domain in practice. It would be great if the crowd sourced data would be released. I also wonder if the paper actually covered enough of the related work in particular with respect to synonym expansion from information retrieval.

Overall, I think the task itself and the resource would be interesting to have at the conference although it's not a radical innovation, hence, I would recommend it for the poster session.

---

### Decision · Program_Chairs · 2019-02-15
**AKBC 2019 Conference Decision**

Accept